# Field test of an active flap system on a full scale wind turbine

Alejandro Gomez Gonzalez[1], Peder B. Enevoldsen[1], Athanasios Barlas[2], and Helge A. Madsen[2]

[1]Siemens Gamesa Renewable Energy A/S, Brande, DK
[2]Technical University of Denmark DTU, Dept. of Wind Energy, Roskilde, DK

**Correspondence:** A Gomez (alejandro.gonzalez@siemensgamesa.com)

**Abstract.**

This article describes a series of validation tests of an active flap system (AFS) on a multi-MW wind turbine. A single blade of a 4 MW turbine with 130m rotor diameter (SWT-4.0-130) is retrofitted in the outer 15-20 m with the AFS. The AFS is controlled remotely with a pneumatic pressure supply system located in the hub of the turbine. The measurements are performed between October 2017 and June 2019 using two different AFS configurations on the blade. A description of the system setup is given, as well as comparisons of measurements and aeroelastic simulations. The measurements quantify the static load control authority of the AFS in atmospheric conditions, providing a preliminary estimate of load impact potential for the concept. The article presents furthermore a new method for the characterization of the load impact of such a system and its dynamic response under atmospheric conditions based on a blade-to-blade load comparison.

## 1 Introduction

Load control and performance optimization of wind turbines using active flap systems has been the subject of numerous studies in the past. It is not within the scope of this paper to give a comprehensive review of the state of the art of active flow control on wind turbines, but to focus solely on the efforts towards active flow control validation at full scale. For an extensive review of the state of the art in smart rotor control, the reader is referred to Barlas (2009), who covers both active control surfaces such as flaps and microtabs, but also other active flow control technologies such as boundary layer control or active twist. Furthermore, the reader is also referred to Pechlivanoglou (2013), who in his dissertation gives a broad overview of different active and passive control technologies for wind turbines as well as comparisons among them, as well as to Johnson (2008) also providing an overview of active load control strategies for wind turbines. Furthermore, a recent technical expert meeting of IEA Task 11 (TEM87, 2017) collects contributions from several groups within the field of active flow control including a variety of technologies such as active flaps (Riemenschneider, 2017; Berg, 2017; Enevoldsen, 2017), plasma actuators (Pereira, 2017), and active leading edges (Holling, 2017), among others.

At wind tunnel scale, Pechlivanoglou (2017) and Samara (2018, 2020) demonstrate the use of active trailing edge flaps on a 3m and 3.5m diameter model wind turbine model at the TU Berlin and the University of Waterloo, respectively. Samara (2020) investigates the control authority of the flap system benchmarking it to the pitch system of the turbine. In these tests, an active flap segment of 20% chord and 22% span coverage was installed on the wind tunnel model. The tests performed reveal that a

flap angle actuation in the range from -20 to +20 deg is similar in terms of load impact to a full section span pitch actuation in the range from 0 to 9 deg.

In terms of full scale validation of active flow control on wind turbines, the academic and industrial contributions are scarce. In the decades prior to the modern boom of wind energy, active flow control on wind turbine blades seemed almost natural, as the technology used for the development of turbine blades was driven mainly by know-how transfer from the aeronautical industry. A clear example of this is the design of a 7.3 MW, 122m rotor diameter turbine within the frame of the MOD-5A program in the early 1980's (MOD5A, 1984). In the design of this turbine, which was never commercialized, three independent ailerons, fully integrated in the outboard section of the blades and activated with a hydraulic system, were used both for loads and power control. Nevertheless, the active flow control strategy of the MOD-5A turbine was not the focus at that time, but rather a choice taken at one of the later design stages as it was believed that this control strategy would be more economically viable than the alternative considered: partial pitch control of the blade. The ailerons were designed as plain hinged flaps extending from approx 60% to 99% of the chord.

In more recent years, a series of tests were performed by RISØ Laboratories in collaboration with Vestas Wind Systems A/S on a Vestas V27-225 kW turbine (13m blade length) equipped with a 70 cm long trailing edge flap as described by Castaignet (2010, 2013a). During the test, which spanned for several hours split among several days, the turbine load control with active flaps was ran in intervals of 2 minutes with, and 2 minutes without the flaps active. During 10 of these measurement intervals, an average blade root flapwise fatigue load reduction of 14% was achieved for a test running 38 minutes using a frequency weighted model predictive control. The controller strategy for this test is further discussed by Castaignet (2013b) and Couchman (2014).

A further full scale test of an active flap system is given described by Berg (2014a, b), where the 9m long CX-100 blade was retrofitted with active flaps integrated in the structure. For this purpose, a rigid hinged flap was incorporated in the trailing edge area of the original blade. The effective hinge area of the flap covered the aft 20% of the chord, of the outboard 20% of the blade span (approx. 2m of the blade). The flap was actuated with a motor embedded in the structure both in constant offsets as well as sinusoidal actuation. During these tests, a significant loss of power production was measured, which is thought to have a relation either to a lack of stiffness of the active flap, an error in blade pitch, or aerodynamic losses due to changes of the aerodynamic performance of the airfoils. The tests also served the dynamic characterization of the system in terms of frequency response and the transient characteristics in terms of time delays.

One relevant (non-flap) active flow control full scale test perfomed on a 1.75 MW wind turbine is described by Matsuda (2017). This test focuses on the use of plasma actuators installed on the leading edge of the blades with the purpose of flow separation control. The tests were performed in swap intervals of 10 minutes and focusing on the effect on the turbine's power production. When the plasma is turned on, a rotor speed increase was measured. For a test period of 22 h distributed over 6 days, an average power increase of 4.9% was measured.

Within the framework of the recent EU project INNWind, an active morphing trailing edge was tested under atmospheric conditions at the Risø campus of the Technical University of Denmark on an outdoor rotating test rig (Ai, 2019; Innwind2.3.3, 2017; Barlas, 2018). The tests conducted herein included both step flap actuation, as well as feed-forward control based on

periodic azimuth variations and also inflow measurements. In contrast to the 2 min cycles of the tests described by Castaignet (2010), these swap tests were performed in 10s cycles over a test period of 5 minutes, repeating this procedure for different operation conditions. The results of these tests showed lift coefficient variation levels in the range of -0.25 to +0.2 (for the step flap tests). Feed forward control tests based on azimuth position and inflow angle showed a reduction of the standard deviations of the flapwise bending moment at the base of the boom of the rotating arm of the rig of 12% and 11%, respectively. In similar tests on the above mentioned rotating test rig described by Madsen (2015a) within the framework of the Induflap1 project, an active flap with 15% chord coverage was tested in flap angle activation steps. A flap angular step magnitude of 5 deg was estimated to be equivalent to a 1 deg full section pitch actuation, similar to the results of Samara (2020).

A good example of the study of active flaps as a potential technology for reduction of LCOE is given in deliverables 2.23 and 2.33 of the InnWind project - see final report Innwind (2017). A significant reduction in blade flapwise fatigue loading is obtained by a combination of individual flap control together with individual pitch control.

The work described in the context of this paper consists of a full-scale long-term validation of two different revisions of an active flap system (described in the next section) on a SWT-4.0-130 turbine. A numerical and experimental characterization of the first revision of the active flap component can be found in (Gomez, 2018). The main purpose of the test is to make an overall aerodynamic and load-wise performance characterization of the system and not to test any specific closed-loop control strategy. Furthermore, the tests presented herein were performed with the goal of characterizing an innovative solution, which in contrast to what has been done in the past, is tested in industrial full scale, during a long period, with remote control and surveillance, as well as with high quality data acquisition of both the turbine and inflow data.

## 2 Test description

The test setup consists of a SWT-4.0-130 turbine (63.4m blade length) retrofitted with the Active Flap System (AFS) on one of the blades, located at the test site Høvsøre, 2 km from the west coast of Denmark. The site is characterized by flat terrain and normally low turbulent flow with a predominant west wind direction. A thorough description of the atmospheric characteristics of the site is given by Peña (2015). The turbine was equipped with a pressure supply system with two pumps working in parallel of type Parker T1-2BL-24-1NEA for the pneumatic activation of the AFS. The air flow was controlled with supply valves in a 3/2 arrangement made up of 2 valves of type SMC VXZ24OFZ2AA 1/2G, one in NC configuration and the second one in NO configuration. Both the pumps and the valves were located in the hub of the turbine. From here, the air was supplied to the AFS via hoses running both along the inside and the outside of the blade up to the start location of the AFS. An internal inflatable hose was responsible for the hinge actuation mechanism at the trailing edge of the airfoil where the AFS was installed. The total power consumption of the system is below 0.5 kW. All systems were designed to be activated remotely interfacing through the turbine communication unit. Two versions of the AFS were tested in independent campaigns as shown in figure 1. The two flap revisions used in phases 1 and 2 are referred to as FT008rev9 and FT008rev10, respectively. Test phase 2 was improved taking into account learnings from phase 1. During phase 2 of testing, besides upgrading the AFS to a newer, more aerodynamically and structural optimized geometry, an upgrade of the pressure supply system was performed focusing on increasing the air

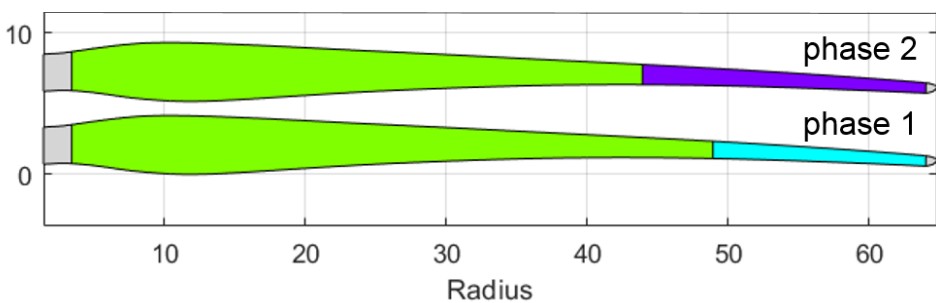

**Figure 1.** Blade layout of AFS

|                          | **Phase 1**            | **Phase 2**                 |
|--------------------------|------------------------|-----------------------------|
| Date                     | Oct 2017 - June 2018   | Dec 2018 - June 2019        |
| Turbine                  | SWT-4.0-130            | SWT-4.0-130                 |
| AFS revision             | FT008rev9              | FT008rev10                  |
| AFS actuation            | discrete positions     | continuous angle variation  |
| Actuation validation type| on-off cycles          | on-off cycles               |
| Location on blade        | 47.5 - 62.5 m          | 42.5 - 62.5 m               |

**Table 1.** Campaign information

storage capacity of the system, as well as a new system allowing continuous variation of pressure levels (in contrast to phase 1,
where only discrete levels are used) - see table 1 for reference. Therefore, the tests presented herein will focus mainly on test
phase 2.

The system installed in the turbine meets the requirements for an accurate measurement of the load handle available by the
use of a particular type of active flap, but has technical limitations for the implementation of high frequency closed-loop control
(e.g. pump capacity, transients related to gas expansion, and viscous losses). Therefore, the tests presented herein are meant as
a technology demonstration, and not as a demonstration of a specific control strategy.

For the validation of the load impact of the AFS, the turbine was instrumented with strain gauges at a distance of 1.2m from
the root of all three blades (configured to measure flapwise and edgewise bending moments) as well as at the tower top position.
Per default, all operational parameters of the turbine such as pitch, rotor speed, and power are continuously logged. The wind
speed and direction, as well as atmospheric pressure, temperature, and humidity, are measured on a met-mast located 2.5D in
front of the turbine. The wind speed and direction at 10 different heights throughout the full extension of the rotor (distributed
between heights of 38m and 155m above ground level) are measured with a nearby Lidar in order to have a good assessment
of shear and veer (see figure 2). The pressure level of the AFS, as well as the digital signals for activation of supply valves are
logged synchronously with all other quantities. The pressure-deflection characteristic of the flap is known from previous wind
tunnel measurements (Gomez, 2018). All quantities are logged continuously for the full measurement period with a sampling

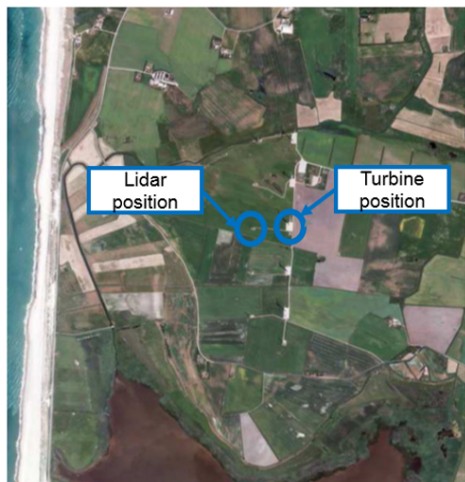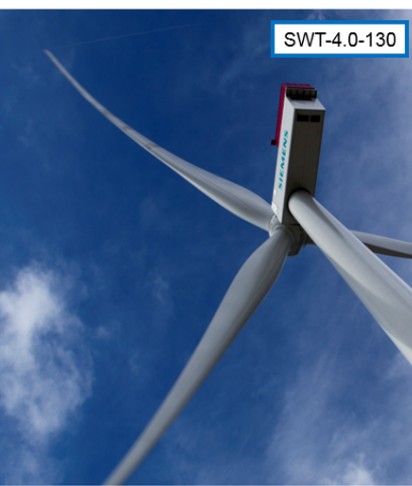

**Figure 2.** Test site layout. Map data taken from GoogleMaps copyright - 2020

frequency of 25 Hz (except the lidar signals, which are sampled with 1 Hz). In addition, the blade is equipped with a remote surveillance camera in order to monitor the AFS (see exemplary view in figure 3 and 4).

The primary characterization of the system is performed by actuating the flaps using pressure steps (equivalent to steps of flap deflection angles), cycling through different pressure levels over constant time intervals. Contrary to the work of Berg (2014a), where the cycles chosen were of 30s, or the work of Castaignet (2013a) with 2 minute cycles, the cycles chosen for the current work are 30 minutes long, allowing both to gather the high frequency transient response of the system, as well as three independent 10-min time stamps of statistical power and load values for every cycle. For standard power and load measurements, the averaging interval of a single measurement point is 10 minutes as this normally covers the turbulence power spectrum of small scale atmospheric turbulence. An interval of 30 minutes captures three sequential measurement values, and allows the system to shift to a different operation condition such that during a full measurement day, the turbine together with the AFS system will be operating numerous times in different modes, avoiding thus that any day-night variations can have an effect on the load comparison.

The cycles were performed during several months, cycling through different flap deflection levels, allowing therefore data acquisition over a very wide range of wind and operating conditions. During the measurement periods, the AFS operated at wind speeds covering the range of approx. $2\frac{m}{s}$ to $15\frac{m}{s}$ for turbulence intensities between approx 3% and 30% as shown in figure 5.

The aerodynamic characteristics of the AFS operated at different pressure levels are known from previous wind tunnel measurements as described by Gomez (2018). An example of the measured normalized aerodynamic coefficients for the first revision of the AFS (FT008rev9) is shown in figure 6 (depicting the impact on lift coefficient and gliding ratio of the airfoil). The curve labeled as *baseline* represents the airfoil without AFS.

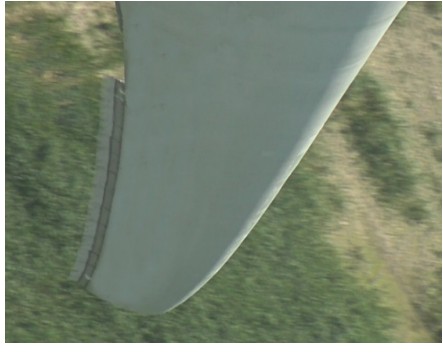

**Figure 3.** Active flap seen from hub

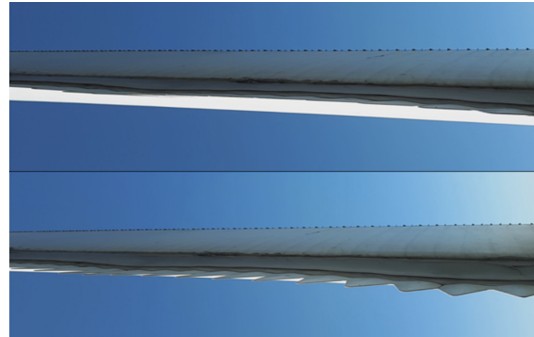

**Figure 4.** View of flap activation

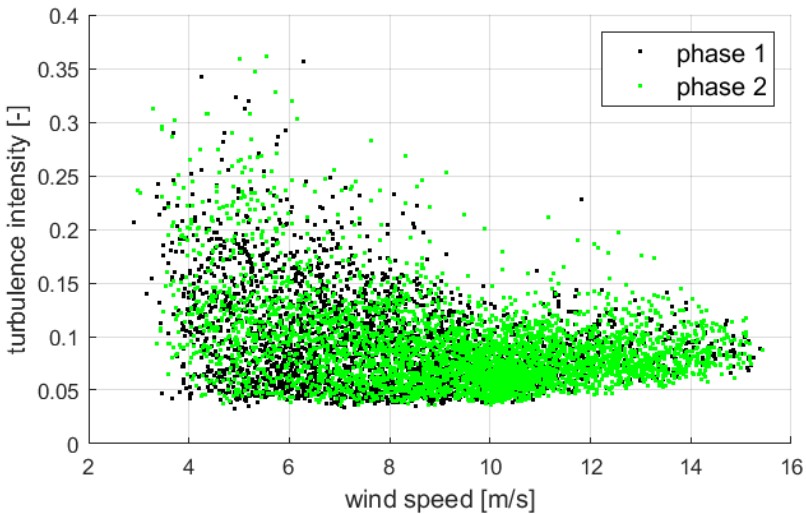

**Figure 5.** Turbulence intensity as a function of wind speed for phase 1 and 2 of the measurement campaign

## 3 Results

This chapter gives a summary of results obtained for both field test campaigns of the AFS. For both campaigns, a series of activation tests were performed to ensure the correct functionality of the system. Visual checks were also performed in order to verify the flap is deflecting properly and is not blocked or restrained in any way.

Due to the stochastic nature of the measurements (with varying levels of inflow turbulence, shear, etc.), the comparison of loads as a function of wind speed is not the best choice. Measured variables as a function of wind speed normally have high scatter due to atmospheric variability, but also due to the coherence effects between the measurement point of the undisturbed wind speed (the met mast), and the location of the turbine. Due to this, a new method for analysis is developed, consisting of a blade-to-blade load variation representation. For this purpose, neighbouring blades are used for the analysis based on flapwise

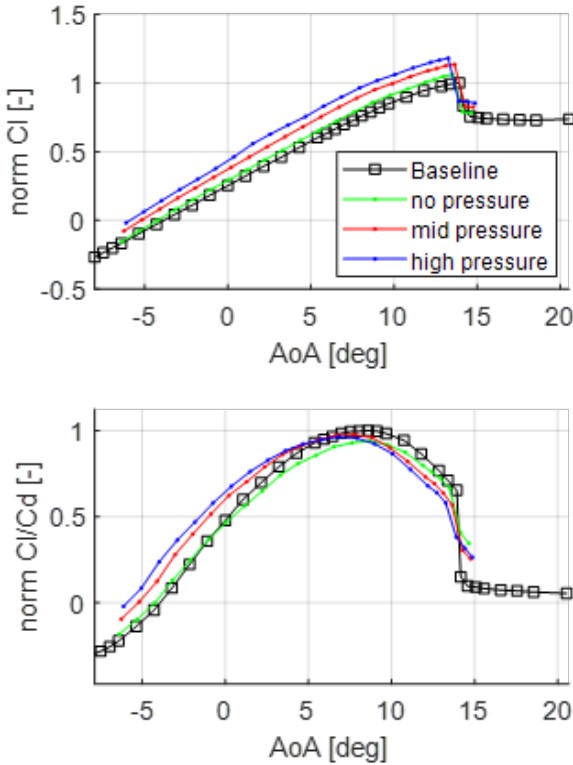

**Figure 6.** Exemplary normalized aerodynamic coefficients for FT008-rev9 AFS

bending moment measurements via strain gauges placed at the root of the blades. Utilizing this method, the uncertainty related
to inflow conditions as well as to coherence in the wind field is removed to a large extent. In what follows, a description of the
method is given.

### 3.1 Blade to blade analysis

The blade-to-blade analysis method (referred to as b2b-method in what follows) consists mainly of two independent types of
analysis: a steady state, and a transient analysis.
The *steady state analysis* is quite straight forward and requires only a turbine equipped with a calibrated means of measuring
bending moments. A standard method for this consists on the calibration of strain measurements in the root area of the blade,
where strain gauges are placed on the intersection points between the contour of the blade and the principal axes of the section.
With independent strain measurements of two different blades (and the corresponding transfer function to obtain bending
moments), the integral load impact of an active device on a blade can be readily measured. One of the blades of the turbine is
150 equipped with the active device in question, whilst the other two blades are left unchanged. In this fashion, the two unchanged

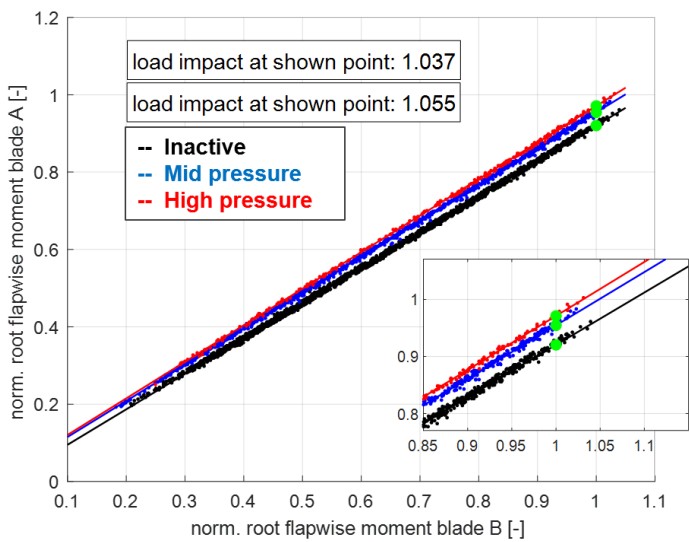

**Figure 7.** Example of a time averaged blade to blade load comparison

blades serve as reference for comparison. This type of comparison between the blade retrofitted with the AFS and the baseline blades has quite low scatter, as both blades are seeing all the time the same time averaged impact of shear, turbulence, veer, and turbine dynamics. Avoiding the use of wind speed measurements increases greatly the correlation of the measured signals because there is no dependence on the coherence of the turbulent wind field. Furthermore, the uncertainty related to single-point wind speed measurements (e.g. related to single point measurements at the met mast for power performance estimation) is removed because the comparison is being done between two load quantities directly, which in their nature are integral quantities (similar to the high correlation which exists between the power signals of two neighbour turbines). A similar method has been previously used by Bak (2016) to evaluate the impact of vortex generators on a full scale wind turbine. An example of such b2b comparison between the blade equipped with the AFS and a reference blade is shown in figure 7, where normalized flapwise bending moments are shown for blades A and B for three different pressure levels for the AFS.

The purpose of the *transient analysis* part of the b2b-method is to extract the transient response (in this case, the step response) of the system as it undergoes changes in time. The complexity of measuring the transient aerodynamic response lies in the fact that this type of high frequency response will normally be hidden in the dynamics of the blades responding to turbulence, vibrations, rotation, etc. The core of the transient analysis of the b2b-method relies on the elimination of the periodic signal dynamics due to rotation and forced vibrations via an artificial time-shift of signals of neighboring blades, together with the elimination of the dynamics due to turbulence via time filtering and ensemble averaging.

The *first step* of the transient b2b-method is to read the bending moment signals from all three blades (the blade retrofitted with the AFS, and the remaining two blades as reference) a sufficient amount of time before and after the dynamic change of the system (e.g. a pressure step of the AFS). Such signals are shown exemplary in figure 8b.

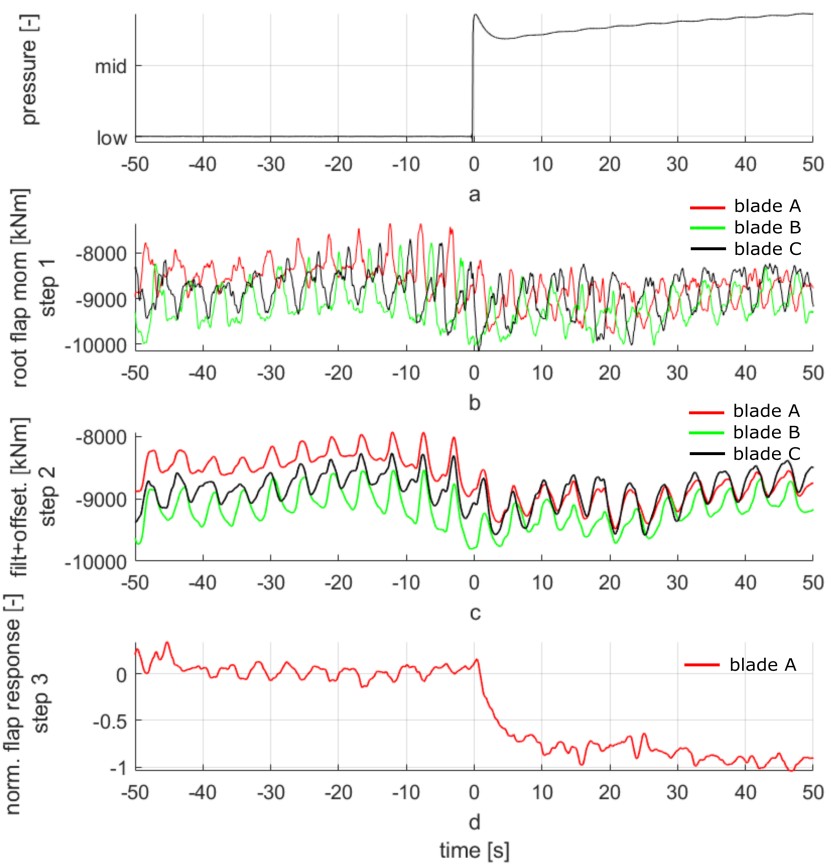

**Figure 8.** Transient signal analysis

The *second step* consists on a low-pass filtering of all signals with an appropriate time constant and a subsequent time shift. In the current analysis, all signals were filtered using a low-pass first order filter with a $1s$ time constant (see figure 8b). The time constant must not be too large, else it has an impact on the dynamics due to the natural phase-shift effect of the filter, and should not be too small else there will be a large amount of high frequency cycles which will still be visible after the ensemble averaging. The leading and trailing baseline blades are shifted backwards and forward in time, respectively, by $\Delta t = \pm \frac{2\pi}{3\bar{\Omega}}$, where $\bar{\Omega}$ is the average rotor speed expressed in $rad/s$ during the respective period. The output of this step is shown exemplary in figure 8c, where in contrast to figure 8b, the signals are now clean of high frequency content and have peaks aligned in time.

The *third step* of the method is to detect the point in time where the dynamic change of the system takes place, and to time-shift the signals such that this time corresponds with $t = 0$ (see example in figure 8a). This will be important for the next step (the ensemble averaging). For the current work, the controller actuation signal of the supply valve of the system

was read directly, knowing then precisely the starting time of the transient response. At this stage, the bending moments of both reference blades are averaged. This average reference value is then subtracted from the bending moment signal of the blade equipped with AFS, and this differential load is normalized based on the amplitude of the load step, leading thus to the normalized impact on loading from this single blade. An example of the result of this stage of the signal processing is shown in figure 8d.

The *fourth and last step* of the transient response extraction is the ensemble averaging of these delta loads across several actuations, such that the stochastic variations are averaged out (e.g. variations due to turbulence). This is shown in the results section in figures 13 and 14.

## 3.2 Field measurements

The first section of results relates to the steady load impact of the AFS used in phase 1 and phase 2 of the field test. The relation
between the flapwise blade root bending moment and wind speed is not monotonic. For modern pitch controlled turbines, the flapwise moment will increase with wind speed until it reaches a maximum close to the region of rated power and in this region the angle of attack (AoA) variations as a function of wind speed are small. Subsequently, the flapwise bending moment will decrease as wind speed further increases, and the AoA of the blade sections will decrease as the blade pitches into the wind. Due to this non-monotonic behaviour of the bending moment and the related changes in AoA, it is important to differentiate
between regions of low and high wind when doing the time averaged analysis. For the measurements referred to here, the data sets were divided into measurement points with 10-min average wind speed below and above $9\frac{m}{s}$, respectively.

All 10-min bins during the measurement periods of phase 1 and 2 were collected and split into low and high wind areas. For this data set, the b2b comparison between blade A (equipped with the AFS) and blade B (one of the baseline blades) is shown in figures 9 through 12. There is a high correlation between these bending moment measurements (seen by the linear relation
and the very low scatter of the plot). Furthermore, the data is filtered based on two different pressure levels representing the operation envelope of the active flap system. The impact of the active system is seen in the fact that two clearly defined lines are seen in these plots. The relative change of loading of blade A vs. blade B at different locations along these lines represents the impact of loading measured at the blade root for different wind speeds. The load levels have been normalized for clarity, where the normalization factor has been chosen as the average peak loading of the bending moment vs. wind speed curve. Therefore,
values close to 1 in these figures represent peak loading of the turbine and are representative of the maximum loading level of the turbine (corresponding to wind speeds close to $9\frac{m}{s}$). For very high operational wind speeds (for example $15\frac{m}{s}$ and above), the bending moment will decrease as already discussed. For this wind speed range, a representative normalized load value of 0.66 is chosen. These two values (1 and 0.66), representative of peak loading and high wind speed loading, respectively, can be seen highlighted in figure 9 through 12 together with the corresponding load ratio.

It can be seen that for phase 1, the load impact at the blade root is measured to be in the range between 3% and 4.2%, whereas for the phase 2, the loading ratio increases up to approx. 5.6% to 10%. It is worth mentioning that the load ratios for outboard locations of the blade which are closer to the AFS will be larger, but these values were not measured during this field

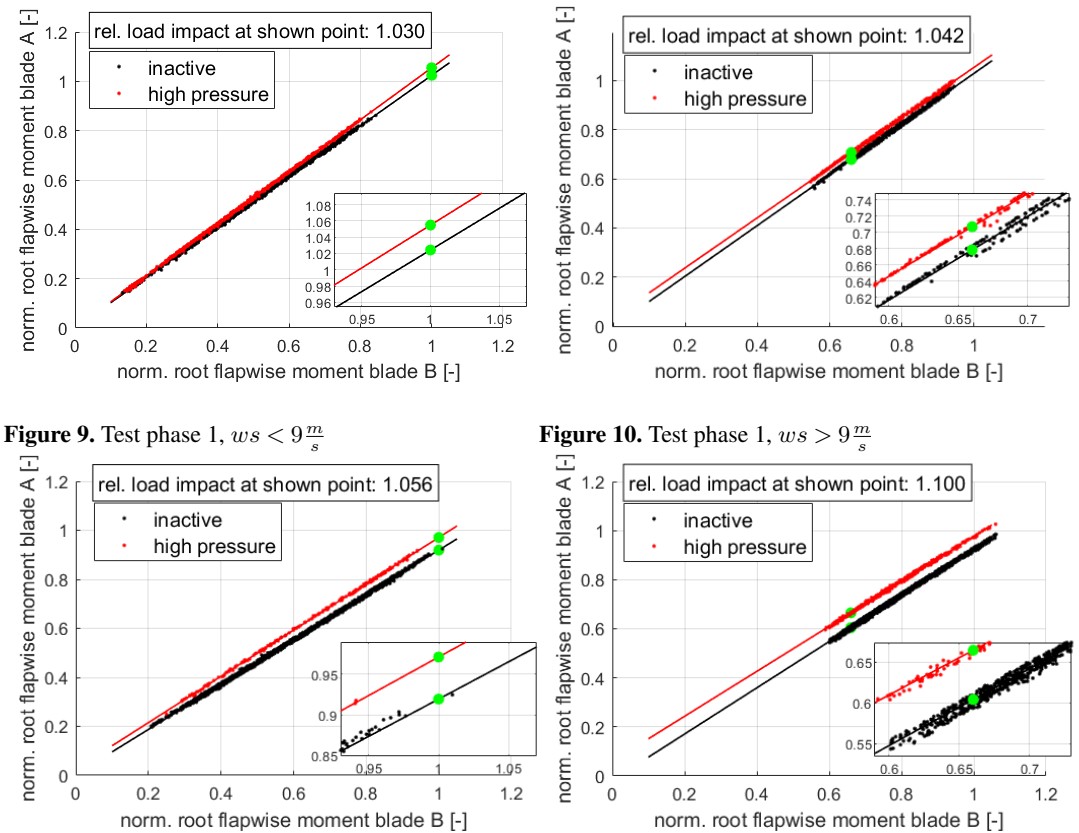

**Figure 9.** Test phase 1, $ws < 9\frac{m}{s}$

**Figure 10.** Test phase 1, $ws > 9\frac{m}{s}$

**Figure 11.** Test phase 2, $ws < 9\frac{m}{s}$

**Figure 12.** Test phase 2, $ws > 9\frac{m}{s}$

test. The load handle of 5% to 10% obtained in phase 2 is one of the major results of the project, highlighting the potential of the AFS to manipulate actively the loads of the turbine.

For the transient analysis of the data accumulated during the field test, the different changes in pressure of the AFS were categorized depending on the start and end value of the pressure level. The data presented herein is given for so-called type H1 and type L2 pressure steps. Type H1 represents a step change in pressure from the lowest to the highest threshold of the system, whereas a type L2 step represents a step change from the highest towards the lowest pressure thresholds. This means in other words that type H1 represent a full activation of the AFS, and type L2 the deactivation. These two type of pressure

jumps show different dynamic behavior due to the differences in flow dynamics during the inflation and deflation of the AFS. The ensemble averaged response of the H1 and L2 pressure steps are depicted in figures 13 and 14. For the analysis shown, the number of individual time series analyzed is given in the plot with the value N. Every dynamic step is extracted at the end / beginning of one of the 30 minute intervals described earlier.

The pressure response measurement of the transient analysis must be used with care due to the physical distance between

the location of the actual measurement, and the location of the AFS. The pressure is measured directly at the exit supply valve.

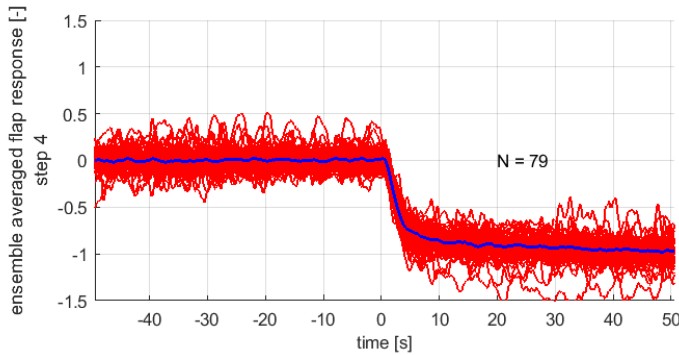

**Figure 13.** Pressure step type H1

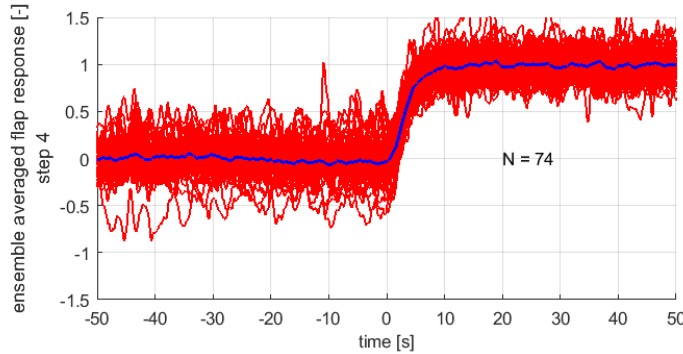

**Figure 14.** Pressure step type L2

This pressure signal is used for filtering the measurements according to the pressure level, but not for the dynamic analysis as it does not represent accurately the instantaneous pressure at the AFS location (due to the flow dynamics in the pressure supply line). The pressure level at the location of the AFS was not measured in these test campaigns due to the long-term duration of the measurements to avoid interference with the lightning protection system. Nevertheless, with the pressure transient at the supply valve location during the activation sequence, it can be seen that the air supply to the AFS reaches saturation (this can be seen from the slow increase in pressure) and will be addressed in future developments.

### 3.3 Aeroelastic simulations

Aeroelastic simulations are performed with SiemensGamesa's in-house solver BHawC see Rubak (2005) and Skjoldan (2011). The simulations are performed in a so-called one-to-one fashion (o2o) (see Enevoldsen (2018) for reference). The o2o calculations are high fidelity aeroelastic simulations performed in a *digital twin* manner, meaning that the structural model is matched exactly to the particular turbine under consideration both in geometry, structural description, and system dynamics, but also where every single 10 min atmospheric inflow measurement point is recreated numerically. The aerodynamic input to the simulation in form of airfoil polars is taken directly from wind tunnel measurements of the AFS. For the full period of phase 1

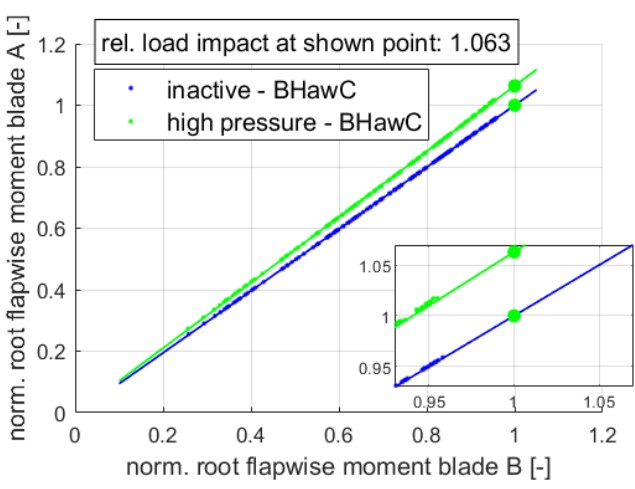

**Figure 15.** Phase 2 simulations at low wind

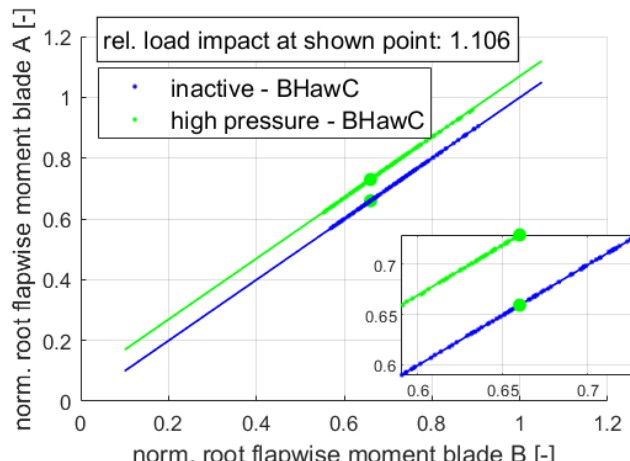

**Figure 16.** Phase 2 simulations at high wind

and phase 2 of the field test there exists therefore the same corresponding period of o2o equivalent aeroleastic simulations. The
input to the simulations is dependent on good inflow measurements and therefore only the wind sector where the metmast /
lidar is unaffected by the wake of any turbine of the park is used for comparison. For this reason, even though the b2b method
is generic for the whole range of directions, only the west sector is taken in consideration for o2o simulations ($270 \pm 30 deg$).

The simulations results presented here focus on phase 2 of the measurement with AFS FT008rev10. In figures 15 and 16,
the time-averaged b2b simulation results for the blade equipped with AFS and a baseline blade are shown. These results,
similarly as for the experimental results, are shown for low wind and high wind regimes below and above $9 \frac{m}{s}$, respectively.
As can be seen from the figure, the aeroelastic simulations are in very good agreement with measurements, predicting a load
impact of 6.3% representative for peak loading, compared to the measured 5.6%, and 10.6% compared to the measured 10.0%
representative for high wind situations (cp. figures 10 and 12).

### 3.4 Discussion of results

Both the aeroelastic simulations as well as the full scale validation of this flap concept show a load handle which allows
to actively adjust the blade root flapwise loading level in the range of 5%-10% depending on the wind speed range. This
load handle comes at the cost of a slight reduction in gliding ratio (see figure 6). Even though a reduction of gliding ratio
would normally be related to a loss of aerodynamic performance, this is only true for "non-active" blades, and affects mainly
the variable speed region of turbine operation. The reason for this is that the aerodynamic performance is also influenced
by the induction levels of the blade, which in turn are highly dependent on the lift level of the airfoils. Depending on the
general induction level of the rotor, the blade design strategy (in terms of induction spanwise distribution), and the operation
characteristics in the constant speed area just before rated power, the power performance of a modern turbine will usually
be more sensitive to variations of lift levels, than to changes of gliding ratios. In this area of the power curve, the ability to

increase lift levels will in general overshadow small penalties of gliding ratios. Following this line of thought, an offshore turbine operating at a site with high mean wind speed and having a wide constant speed region prior to rated power, will be less sensitive to gliding ratio penalties and will benefit more from the ability to actively change lift levels, than a smaller onshore turbine at a low wind speed site with a wide variable speed range.

A potential load handle of 5-10% is certainly interesting from a wind turbine design perspective. Such a load handle can be used in two ways: the first one is during the design phase of a new platform in order to enable a more cost effective dimensioning of the main components, and the second one is to upscale the rotor of an existing platform while maintaining the loads within the allowable envelope, having thus a positive impact on LCOE - see e.g. Barlas (2016).

## 4 Conclusions

Two independent long term validation campaigns were conducted for a pneumatic active flap system on a full scale wind turbine. The AFS was actuated in an on-off fashion in order to assess the load impact of the system. The first revision of the AFS was activated at discretely varying pressure levels and showed a potential load impact between 3-4% for root flapwise bending moments. In a second phase of the testing campaign, the AFS was optimized both aerodynamically and structurally, in combination with an upgrade of the pressure supply system which enabled continuously varying angle activation. This revision of the AFS showed a potential load impact between 5-10% at the root of the blade.

The measurements performed were accompanied with high fidelity aeroelastic simulations where the aerodynamic input was based on polars measured in a wind tunnel, and the atmospheric inflow was based on metmast and lidar measurements. Very good agreement was found between the measurements and the simulation results.

The AFS was shown to operate in a robust way and without major drawbacks. The AFS revisions were tested independently for over 6 months each, covering therefore a wide range of wind speeds, temperatures, and atmospheric conditions. The loads of the turbine were measured as well in a robust and accurate manner, enabling a good estimate of the potential load impact levels of the AFS. Finally, a novel method was developed in order to directly measure the transient behaviour of the AFS in a highly time varying environment.

*Author contributions.* All authors contributed significantly to the work presented within this paper.

*Competing interests.* The authors declare they have no competing interests.

*Acknowledgements.* The work presented in this article is part of the Induflap2 project (Induflap2) which is a research and development project carried out in the period from 2015 to 2019. The authors wish to acknowledge EUDP for the funding given to the project (journal nr. 64015-0069).

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
