# Peer review of "Field test of an active flap system on a full scale wind turbine"

_Wind Energy Science, 2020_

## Referee Comment (RC1) · Anonymous Referee #1 · 19 Jun 2020

The article concerns a field test and an aeroelastic investigation of the use flaps on a full-scale wind turbine. The article is interesting in that this is first results of an investigating of using flaps on a full-scale MW wind turbine, and as such it deserves publication. However, several issues needs to be ameliorated before the present paper can be published.

- I understand that it is not the purpose of the article to make a full literature review of the subject. However, this sound more like an excuse for not digging more into the pertinent literature. Nearly all reference and cited papers are related to own previous works. To justify publication, it is required more explicitly to argue the originality of the work and how it is inscribed in other literature and investigations.
- In the paper we learn that there is an effect of using active flaps, and that this effect indeed can be measured through the flapwise root moment. However, as we can see on Fig. 4 there is also a price to pay with respect to aerodynamic efficiency at high angles of attack. I think this balance between enhanced aeroelastic features vs the changed aerodynamics should be discussed. Furthermore, a general discussion of the motivation for using the flaps and a discussion of the potential gain of the achieved results are required, like, is a potential load impact of 5-10 % worthwhile of going after?
- Legends explaining the colors in Fig.6 are missing.
- Besides being limited to own research work, the citations and reference list is incomplete. On page 11 there are references to Fisker (2011) and Enevoldsen (2018) that do not appear in the reference list.

---

## Referee Comment (RC2) · Anonymous Referee #2 · 15 Jul 2020

**Journal:** WES
**MS No.:** wes-2020-21
**MS Type:** Research articles
**Title:** "Field test of an active flap system on a full scale wind turbine"
**Author(s):** Alejandro Gomez Gonzalez, Peder B. Enevoldsen, Athanasios Barlas, and Helge A. Madsen

**General comments:**

An innovative Active Flap control System (AFS) is evaluated on field tests of a multi-MW wind turbine in an open-loop configuration. The tests were performed during long cycles (30 min) and for several months. The given objective of the authors is to control loads. The authority is evaluated during this period from root flapwise moments using strain -gauges installed at the blade root. A blade to blade method is used, consisting on comparing the loads between blades with only one equipped with the AFS. Results demonstrate the control efficiency with a global root load gain from 5% to 10%. Also, from control step functions and a specific post-processing method, the control dynamics is evaluated. It can be approximated to a first order system with a time response of the order of few seconds. Finally, aero-elastic simulations were performed and were able to reproduced the load gain of the Active Flap control System.

As pointed out by the authors, academic and industrial contributions on active flow control on full scale wind turbines are scarce. In that sense, this work is remarkable.

However, important descriptions are missing and should be tackled before the publication of this paper. They are listed below:

**Point 1**- The control objective is not clear as the time response of the control system do not allow a load control in closed-loop.
**Point 2**- Details on the inflow characterization are missing to evaluate the representation of the samples used relatively to the atmosphere conditions on the given terrain.
**Point 3**- Details on the actuator set-up are also missing. We don't know how the system is working.
**Point 4**- Some descriptions on the evaluation of the actuation impact are not clear

**Point 1:  Control objective ?**

The blade to blade analysis method is very interesting and certainly the best method to easily evaluate the authority of a new control system. However, the needed control authority dependents on the objective of the control. Such authority is certainly not necessary for alleviating loads, while this is certainly unavoidable to control the output power of the turbine (similarly as the pitch control). Also, for alleviating loads fluctuations, due to the blade passages in front of the wind turbine mast for instance (the famous tower shadow effect), the control system must be faster than the 3p frequency which can be as fast as to 0.5s. From the transient analysis of the present study, the system developed is too slow to alleviate the load fluctuation from tower shadow effect for instance.

**Question 1:** It is therefore important to have a literature review on the different control objectives in the field of wind energy (power control , load and fatigue …) and the associated time scales (from large atmospheric boundary layer scales to small shear layer vortices or 3P ...). Authors should specify the objective study within this literature review.  Also, in this review, it is certainly worth mentioning the only AFC using fluidic actuator (plasma actuator) work on a multi-MW

turbine (Matsuda et al 2017), able to perform a very fast actuation (10kHz).

H. Matsuda, M. Tanaka2, T. Osako2, K. Yamazaki1, N. Shimura, M. Asayama and Y. Oryu "Plasma actuation effect on a MW class wind turbine" International Journal of Gas Turbine, Propulsion and Power Systems, Feburary 2017, Volume 9, Number 1

**Point 2: Details on the inflow characterization:**

In order to evaluate if the samples used include sufficient atmosphere conditions, on must know the atmosphere conditions of the terrain and the terrain topology. More details are therefore needed for this:

**Question 2:**
p3L75: instead of giving the wind speed and turbulent intensity range( 2m/s → 15 m/s  Ti = 3% → 30%), please give the wind roses for the the wind speed and the Turbulent intensity used in the study.

**Question 3:**
P3L73: " the cycles were performed during several month ..."
Please be more specific, how many month, which month ?

**Question 4:** what is the type of Sensor and their acquisition frequency ?
Can you evaluate the atmosphere stability with these sensors for instance ?

**Question 5:**
Where are the 10 heights measurements (including the topology of the terrain) ?

**Point 3: Details on the actuator set-up**

The description of the actuator system is never explicitly given. We don't know actually how the actuator is working unless we go the the publication from Gonzales et al (2028).

A Gomez Gonzalez, P B Enevoldsen, B Akay, T K Barlas, A Fischer, H Aa Madsen Experimental and numerical validation of active flap for wind turbine blades. Journal of Physics: Conf. Series 1037 (2018) 1234567890  ''"" 022039

This is particularly annoying to evaluate if the targeted objective ( load control) is reachable. The needed time response of the system for load control (due to shadow effect) seems to be not reachable from the present system. Delays come from the whole control system arrangement itself, one order of magnitude slower. Indeed, this can't be the time response of aerodynamic loads which is faster, of the order of 0.2s (for 1.25m blade chord and a relative velocity of 60m/s). Moreover, this delay does not include the whole system:
P10L172:    "The pressure response measurement of the transient analysis must be used with care due to the physical distance between the location of the actual measurement, and the location of the AFS. The pressure is measured directly at the exit supply valve."

**Question 6:** A more detail description of the actuation system with the tubing arrangement (including the valve type, dynamic characteristics) should be included to at least evaluate what objective can be reached by this control system.

**Question 7:** It seems that the control system is mostly interesting for power control. In that case, it

is important to evaluate the net benefit.
Can you provide more details on at least the power supply needed to compress the air ?
At maximum the impact of the additional weight and the impact on the rotor imbalance should be provided.

**Point 4: Evaluation of the actuator impact:**

p6L95: "A standard method for this consist on the calibration of strain measurements in the root area of the blade, where strain gauges are placed on the intersection points between the contour of the blade and the principal axes of the section. With independent strain measurements of two different blades (and the corresponding transfer function to obtain bending moments), the integral load impact of an active device on a blade can be readily measured."

**Question 8:** The location of the strain gauges are not clear: which section is used ? Principal axis are not given. What transfer function are you using ? Are the bending moments known from another source ? What is the calibration procedure used in the present study ?

**Question 9:** Also, the calibration of strain gauges on operating wind turbines is generally a long/heavy/costly and not accurate procedure. Is the evaluation of the bending moment necessary in the present blade to blade analysis ? Can't we use the strain gauge measure directly ?

P6L104: "Furthermore, the uncertainty related to point-wise wind speed measurements is removed."

Yes, but this is valid only if the statistical converge is reached. Regarding the atmosphere changes, there is diurnal changes, seasonal changes, dependence on the terrain etc …

**Question 10:** You should moderate this sentence, the uncertainty related to point-wise wind speed measurements is only smoothed and valid for a limited range in the atmosphere/terrain conditions (which are not given in the paper).

**Question 11:** There is certainly limitations that are linked to the statistical convergence of data, which is certainly dependent on the coherence of the turbulent wind field contrary to what is said in the text. In other terms, why 30 min and not 1hour, 2hours, 1 day … for your statistics ?
Have you looked on how the statistics converge towards the final value ?

**Minor correction:**
p13 237: Skjoldan, P.F. while  Fisker in p11L181

---

## Author Comment (AC1) · 9 Aug 2020

Reply to comments of RC1 and RC2

In the case that similar comments were done, I have grouped my reply to both.

| Comments | Reply |
|---|---|
| • I understand that it is not the purpose of the article to make a full literature review of the subject. However, this sound more like an excuse for not digging more into the pertinent literature. Nearly all reference and cited papers are related to own previous works. To justify publication, it is required more explicitly to argue the originality of the work and how it is inscribed in other literature and investigations.

Question 1: It is therefore important to have a literature review on the different control objectives in the field of wind energy (power control , load and fatigue …) and the associated time scales (from large atmospheric boundary layer scales to small shear layer vortices or 3P ...). Authors should specify the objective study within this literature review. Also, in this review, it is certainly worth mentioning the only AFC using fluidic actuator (plasma actuator) work on a multi-MW turbine (Matsuda et al 2017), able to perform a very fast actuation (10kHz).

H. Matsuda, M. Tanaka2, T. Osako2, K. Yamazaki1, N. Shimura, M. Asayama and Y. Oryu "Plasma actuation effect on a MW class wind turbine" International Journal of Gas Turbine, Propulsion and Power Systems, Feburary 2017, Volume 9, Number 1 | Thank you for your comment, and I fully understand the concern. To keep the paper concise, the literature review focused initially solely on field experimental work on active flaps, which indeed is quite a limited field.

In the revised version, I will include further the references, in particular to include the experimental wind tunnel work of Pechlivanoglou (TU Berlin), field work performed at the DTU on morphing flaps on a rotating test rig as a part of the InnWind project, and experimental work with trailing edge flaps of Samara and Johnson.

As suggested by referee nr. 2, I will include the work of Matsuda et al, which I was aware of, but intendedly did not include it in the first revision of the paper as the focus is on plasma actuators and not on trailing edge flaps. In the updated revision of the paper I will include as it fits well with the experimental character of our submission

Regarding the literature review of different control objectives as suggested by referee nr. 2, I think this is out of the scope of this paper. Control objectives for active devices on wind turbines is a very vast subject, both from the controller objective perspective, but also regarding the different devices (SJA, plasma, active gurneys, TE flaps, active leading edge, spoilers, blowing, suction, etc. It is the opinion of the authors that his field is already so wide, that a publication of experimental demonstration character such as the one we are proposing should not try to cover the literature of controller objectives.

It is important for the authors to highlight, that the purpose of the work shown in our publication and also the work which was presented at the conference is focused on an experimental demonstration and testing of a system, but not the test of a particular controller strategy. |

| | |
|---|---|
| • In the paper we learn that there is an effect of using active flaps, and that this effect indeed can be measured through the flapwise root moment. However, as we can see on Fig. 4 there is also a price to pay with respect to aerodynamic efficiency at high angles of attack. I think this balance between enhanced aeroelastic features vs the changed aerodynamics should be discussed. Furthermore, a general discussion of the motivation for using the flaps and a discussion of the potential gain of the achieved results are required, like, is a potential load impact of 5-10 % worthwhile of going after? | The discussion of lower aerodynamic efficiency is for sure a very interesting one. The full aero efficiency of the turbine can not be boiled down to the gliding ratio only, but depends also on the mean induction levels of the blade, the blade design strategy, and the operation strategy in the region around the shoulder of the power curve (where the performance is more sensitive to lift levels than to the lift to drag ratio. I will comment this balance of load handling vs. aerodynamic performance in a concise manner. To give an example, in the case of modern large offshore blades where the outboard area is designed towards low induction in order to allow for platform upscaling and where the AEP is geared towards high mean wind speeds, a penalty in lift over drag is over shadowed by the ability of increasing the induction level via lift levels. On the contrary, a smaller onshore turbine where the blades are designed for operation at lower mean wind speeds, the penalty of the lower gliding ratio will be more significant and may overshadow the ability to have the desired control authority from the active flaps. This is just to say that this is not a black or white discussion 😊

Having a potential load handle of 5-10% is indeed worth going for and this was not clearly stated in the paper. From an industry perspective, load reductions can not be translated into LCOE in a direct manner on an existing platform and a cost-out of an existing turbine would not be the correct way to go, as the overhead costs would overshadow the improvement in LCOE. Therefore, such a load handle can be used in two ways: the first one is during the design phase of a new platform in order to enable a more cost effective dimensioning of the main components, and the second one (which is economically more attractive) is to upscale the rotor and maintaining the load envelope à this option has the higher impact on LCOE. |
| • Legends explaining the colors in Fig.6 are missing. | I will add these. In the plots with more than one time series (the two middle plots), each of them corresponds to the flapwise bending moment of one of the blades A, B, and C, and in the last plot, the loading corresponds to that of blade A after having applied the blade to blade comparison method (i.e. relative to blades B and C). These legends will be added |

[Figure]

| | |
|---|---|
| • Besides being limited to own research work, the citations and reference list is incomplete. On page 11 there are references to Fisker (2011) and Enevoldsen (2018) that do not appear in the reference list. | The comment to the references is addressed in the first point of the reply.

The references to Enevoldsen and Fisker are included, but I can see that there are two typos. In the reference to Peder Enevoldsen I wrote 2014 instead of 2018. In the refernce to Peter Skjoldan Fisker I used his middle name (Fisker) instead of his last name (Skjoldan) when referencing it in the text. I will correct this. |

Enevoldsen, P.B.: Load validation and advanced modelling, Advances in Rotor Blades for Wind Turbines - IQPC Conference, April 24-26 2014. Bremen, Germany, 2014.

Energiteknologiske Udviklings-. of Demonstrationsprogram. Project Journal Nr. 64015-0069.

Skjoldan, P.F.: Aeroelastic modal dynamics of wind turbines including anisotropic effects. Ph.D. Thesis, DTU Risoe-PhD-66, 2011.

**Point 2: Details on the inflow characterization:**

In order to evaluate if the samples used include sufficient atmosphere conditions, on must know the atmosphere conditions of the terrain and the terrain topology. More details are therefore needed for this:

**Question 2:**
p3L75: instead of giving the wind speed and turbulent intensity range( 2m/s → 15 m/s  Ti = 3% → 30%), please give the wind roses for the the wind speed and the Turbulent intensity used in the study.

**Question 3:**
P3L73: " the cycles were performed during several month ..."
Please be more specific, how many month, which month ?

**Question 4:** what is the type of Sensor and their acquisition frequency ?
Can you evaluate the atmosphere stability with these sensors for instance ?

**Question 5:**
Where are the 10 heights measurements (including the topology of the terrain) ?

Reply to question 2: The wind rose is not given because it would the secto

Reply to question 3: This is specified in Table 1 (see below)

| | Phase 1 | Phase 2 |
|---|---|---|
| Date | Oct 2017 - June 2018 | Dec 2018 - June 2019 |
| Turbine | SWT-4.0-130 | SWT-4.0-130 |
| AFS revision | FT008rev9 | FT008rev10 |
| AFS actuation | discrete positions | continuous angle variation |
| Actuation validation type | on-off cycles | on-off cycles |
| Location on blade | 47.5 - 62.5 m | 42.5 - 62.5 m |

**Table 1.** Campaign information

Reply to question 4: The sensor signals of the met-mast are provided by an external supplier: in this case SGRE has a contract with the DTU dependency at the Høvsøre test center as responsible party for the calibration of meteorological instruments and signal availability. Due to this, the sensor manufacturer is not known. Nevertheless, it is the same type of instrumentation that at SGRE normally is used for power performance measurements compliant with IEC61400-12 and the sensors are compliant with the norm requirements.

The acquisition frequency is 25Hz for the instruments in the metmast and 1Hz for the lidar signals, this is mentioned in page 3 line 65.

With regards to atmospheric stability, this is normally the case when no Lidar data is available, and the atmospheric stability is then estimated via the Monin-Obukhov length. In the current measurement setup, the direct profile is measured, and the additional information of the mixing length parameter for stability analysis is not relevant.

Reply to question 5: The measurement heights are (measured above ground): 38, 47, 59, 71, 83, 95, 107, 131, 143, and 155m

The site is flat (ie. It is flat in accordance to the requirements of table B.1 of annex B of IEC61400-12-1). A reference to a report on 10 year boundary layer meteorology at Høvsøre will be included in the paper.

If the terrain complies with the requirements of Table B.1, then no site calibration is required.

If the terrain characteristics are within an additional 50 % of the limits of the maximum slopes shown in Table B.1, then a flow model can be used to determine if a site calibration measurement can be avoided. The flow model shall be validated for the type of terrain. If the flow model shows a difference in wind speed between the anemometer position and the turbine's hub less than 1 % at 10 m/s for the measurement sectors, then no site calibration measurement is required.

Otherwise a site calibration measurement is required.

**Table B.1 – Test site requirements: topographical variations**

| Distance | Sector | Maximum slope % | Maximum terrain variation from plane |
|---|---|---|---|
| <2 *L* | 360° | <3* | <0,04 *(H+D)* |
| ≥2 *L* and < 4 *L* | Measurement sector | <5* | <0,08 *(H+D)* |
| ≥2 *L* and <4 *L* | Outside measurement sector | <10** | Not applicable |
| ≥4 *L* and <8 *L* | Measurement sector | <10* | <0,13 *(H+D)* |

\*   The maximum slope of the plane, which provides the best fit to the sectoral terrain and passes through the tower base.

\*\*  The line of steepest slope that connects the tower base to individual terrain points within the sector.

[Figure]

IEC  2037/05

**Point 3: Details on the actuator set-up**

The description of the actuator system is never explicitly given. We don't know actually how the actuator is working unless we go the the publication from Gonzales et al (2028).

A Gomez Gonzalez, P B Enevoldsen, B Akay, T K Barlas, A Fischer, H Aa Madsen Experimental and numerical validation of active flap for wind turbine blades. Journal of Physics: Conf. Series 1037 (2018) 1234567890 ''"" 022039

This is particularly annoying to evaluate if the targeted objective ( load control) is reachable. The needed time response of the system for load control (due to shadow effect) seems to be not reachable from the present system. Delays come from the whole control system arrangement itself, one order of magnitude slower. Indeed, this can't be the time response of aerodynamic loads which is faster, of the order of 0.2s (for 1.25m blade chord and a relative velocity of 60m/s). Moreover, this delay does not include the whole system:
P10L172:  "The pressure response measurement of the transient analysis must be used with care due to the physical distance between the location of the actual measurement, and the location of the AFS. The pressure is measured directly at the exit supply valve."

**Question 6:** A more detail description of the actuation system with the tubing arrangement (including the valve type, dynamic characteristics) should be included to at least evaluate what objective can be reached by this control system.

The general comments of point 3 require further clarification from our side. The purpose of the test was to do a technology demonstration at full-scale in order to start discovering the limitations of the systems and to perform a technology exploration. It was never the objective to test a particular controller strategy (this was also mentioned several times during the presentation at the conference). I will make this more clear in the paper.

Therefore, the system is currently responding according to the physical limitations of the setup, including the pump capacity for air supply, as well as the gas dynamics of flow compression and viscous losses in the supply pipes (which are located inside the blade as the sketch below shows.

Therefore, we are not proposing that the current system is meant for load alleviation in its current state due to the time response. This also led to a very nice discussion during and after the conference presentation. Here I emphasized again, that the aim of the project was to design, install and test a technology demonstrator where we could be able to measure in full scale the available load handles, and further develop the methods required for measuring these with good accuracy (which was the blade-2-blade comparison method developed and described in the paper)

Reply to question 6: Yes, due to conciseness and page limitation, the reader is referred to the publication mentioned. All pneumatic components are placed in the hub. Air is supplied with a two pumps working in parallel of type Parker T1-2BL-24-1NEA. The valves used for control are in a 3/2 arrangement made up of 2 valves of type SMC VXZ24OFZ2AA 1/2", one in NC configuration and the second one in NO configuration. This will be mentioned in the paper.

[Figure]

**Question 7:** It seems that the control system is mostly interesting for power control. In that case, it

is important to evaluate the net benefit.
Can you provide more details on at least the power supply needed to compress the air ?
At maximum the impact of the additional weight and the impact on the rotor imbalance should be provided.

Reply to question 7: Similar as my reply to comments to point 3 above, the aim of the paper is not to demonstrate any control strategy, but to demonstrate a technology in full scale and measure the potential load impact. The system uses less than 0.5 kW at full operation pump speed, therefore, due to the low value, no further considerations to power consumption were made.

Rotor imbalance is not an issue. This demonstrator was installed with the AFS on a single blade with the purpose of subsequently being able to estimate the load potential with help of the blade-2-blade comparison method (which had not been possible with a full 3 blade installation). A real system would certainly have a 3 blade full blade installation, and therefore rotor imbalance would not play a role.

| | |
|---|---|
| **Point 4: Evaluation of the actuator impact:**

p6L95: "A standard method for this consist on the calibration of strain measurements in the root area of the blade, where strain gauges are placed on the intersection points between the contour of the blade and the principal axes of the section. With independent strain measurements of two different blades (and the corresponding transfer function to obtain bending moments), the integral load impact of an active device on a blade can be readily measured."

**Question 8:** The location of the strain gauges are not clear: which section is used ? Principal axis are not given. What transfer function are you using ? Are the bending moments known from another source ? What is the calibration procedure used in the present study ?

**Question 9:** Also, the calibration of strain gauges on operating wind turbines is generally a long/heavy/costly and not accurate procedure. Is the evaluation of the bending moment necessary in the present blade to blade analysis ? Can't we use the strain gauge measure directly ? | Reply to Questions 8 & 9
The strain gauges are located at 1.2m from the blade root at the intersections of the principal axis of that section with the blade contour. The geometrical information of the blade contour is SGRE proprietary and can therefore not be disclosed. It will be made a mention in the paper regarding the spanwise location of the strain gauges.

The calibration procedure of strain gauges in the blades is quite straight forward and is performed for each one and every prototype turbine. These procedures are well known in the industry and some suggestions are also given in IEC 61400-13 (see screenshot below). For the present case, the calibration is performed is gravity based as described in section 4.2.2.2 of the standard aforementioned. |

**4.2.2 Measurement of blade root bending moments**

**4.2.2.1 Instrumentation**

Flap and lead-lag bending moments shall be measured. For lightning and environmental protection, it is recommended that the sensors be mounted within the blades rather than on the outer surface, where convenient. This will also lead to better protection during handling.

Strain gauges should be applied in such a position that cross-sensitivities between lead-lag and flap-load measurements are minimized. Applying the gauges to a part of the blade root which is as nearly cylindrical as possible may facilitate this. Regardless of the mounting location, cross-sensitivity should be measured and corrected according to B.2.1.

In the case of pitch-regulated turbines, the above advice also applies. However, for consistency, the sets of gauges should be oriented so that they are parallel to and perpendicular to the chord line at 70 % radius.

**4.2.2.2 Calibration**

The blade-root load sensors should be calibrated by external force application close to the blade tip. Alternatively, the signals for the lead-lag and flap-bending moment in the blade root can be calibrated using the blade mass as a calibration load in case the blade can be pitched over at least 90°. Since the load signals are designed to measure the bending moments in the blade root at the position of the strain gauges, the calibration has to be performed using the values of the mass and centre of gravity of the part of the blade outside the strain-gauge position for the determination of the calibration load. This requires exact knowledge of the distribution of the blade mass per unit length along the blade axis.

It should be noted that using the blade mass to calibrate the loads might limit the calibration load range and result in a higher calibration uncertainty.

**4.2.2.3 Calibration check**

By rotating the rotor slowly around 360°, the blade mass causes a variation in the lead-lag signal. If pitching is possible, the variation in the flap wise signal can also be measured. The variations have to be measured at initial calibration in order to determine the reference for repeated checks later on. This check shall be done at low wind speeds. When checking the lead-lag moments, it is recommended to yaw the wind turbine 90° in relation to the wind direction.
* * *
P6L104: "Furthermore, the uncertainty related to point-wise wind speed measurements is removed."

Yes, but this is valid only if the statistical converge is reached. Regarding the atmosphere changes, there is diurnal changes, seasonal changes, dependence on the terrain etc …

**Question 10:** You should moderate this sentence, the uncertainty related to point-wise wind speed measurements is only smoothed and valid for a limited range in the atmosphere/terrain conditions (which are not given in the paper).

Reply to comment and to question 10: The blade-2-blade comparison method precisely addresses the diurnal and seasonal changes that you mention, because the AFS-blade and the other two reference blades are seeing the same inflow during the same 10-min interval, that is the nice thing about the analysis. It is analogue to side-by-side analysis of turbine performance when performed according to IEC. I will include this in the paper
* * *
**Question 11:** There is certainly limitations that are linked to the statistical convergence of data, which is certainly dependent on the coherence of the turbulent wind field contrary to what is said in the text. In other terms, why 30 min and not 1hour, 2hours, 1 day … for your statistics ?
Have you looked on how the statistics converge towards the final value ?

Reply to question 11: For standard power and load measurements, intervals of 10 minutes are used as this normally covers the turbulence power spectrum of small scale atmospheric turbulence (see Van der Hoven spectrum below). 1 day is not chosen because you would have a

clear day-night cycle in the data set. 1 hour or 2 hours could have also been options. We intended to gather a higher number of transients, therefore we chose 30 minutes instead of periods of 1 or 2 hours. We chose 30 minutes, and not 10 minutes (which else would have been standard), in order to avoid having a significant impact of the transient behavior on the steady state values. I will comment this in the paper.

[Figure]

**Figure 2.2:** Van der Hoven spectrum (1957) as drawn by Alan Davenport (Isyumov, 2012).

---

## Referee Report (RR1)

**Journal:** WES
**MS No.:** wes-2020-21
**MS Type:** Research articles
**Title:** "Field test of an active flap system on a full scale wind turbine"
**Author(s):** Alejandro Gomez Gonzalez, Peder B. Enevoldsen, Athanasios Barlas, and Helge A. Madsen

**General comments:**

Thank you for your corrections and for the additional informations. This work is now well suitable for publication from my point of view. See below the detailed comments.

**Point 1**- The control objective is now clear to me thanks to the added references. It can be classified as a low frequency load alleviation system (flapwise fatigue loads) when combined with pitch control.

**Point 2**- The added description of the terrain together with the cited reference and the added turbulent intensity information are found sufficient to describe the wind inflow conditions for this first control demonstration.

**Point 3**- The actuator set-up description and limitation are now clear.
Just a small comment:
- L85: what NC and NO stand for ?

**Point 4-**
**Question 10:** I disagree with the authors, the blade-2-blade comparison method is indeed a good method to evaluate the efficiency of the control, but for a given external working conditions of the turbine. For instance, the control efficiency might be different with an increased turbulent intensity environment. This could be easily checked now by classifying the control efficiency version the turbulent intensity level of the atmosphere (the efficiency being characterized by the distance between your reference case 'inactive' and the control case 'Mid pressure' or 'High pressure'). In fact, this might be a very interesting information to add without much additional effort.

Similarly, this could be done with diurnal, seasonal changes or specific external conditions such as rain, snow … etc. However, I understand that this is a first step study for this new system and I don't ask for additional field experiments for this paper.

---

## Author Response (AR3)

Reply to comments of RC1 and RC2

In the case that similar comments were done, I have grouped my reply to both.

| Comments | Reply |
|---|---|
| Referee report nr.1 Point 3

The referee asks for clarification of the abbreviations NO and NC | NO and NC are abbreviations commonly used when specifying valve types, and stand for "normally open" and "normally closed", respectively.

A magnetic valve of NO type is therefore always open, unless a current input is given (then it closes). On the contrary, a NC valve is always closed, and closes upon a current signal.

This has been updated in the document |

[revised manuscript text omitted]